# Comparison of Perinatal Outcome of Delta and Omicron Variant of COVID-19 Infection—A Retrospective Observational Study

**DOI:** 10.3390/medicina60060935

**Published:** 2024-06-03

**Authors:** Natasa Karadzov Orlic, Vesna Mandic-Markovic, Svetlana Jankovic, Relja Lukic, Zagorka Milovanovic, Dragana Maglic, Dunja Popov, Marko Stankovic, Suzana Drobnjak, Dasa Preradovic, Zeljko Mikovic

**Affiliations:** 1School of Medicine, University of Belgrade, 11000 Belgrade, Serbia; vesna.m.mandic@gmail.com (V.M.-M.); svetlanajankovic.r@gmail.com (S.J.); reljalu@gmail.com (R.L.); zaga_mil@yahoo.co.uk (Z.M.); dragana.maglic@gmail.com (D.M.); mikovic.zeljko@gmail.com (Z.M.); 2High-Risk Pregnancy Unit, Obstetrics/Gynecology Clinic “NarodniFront”, 11000 Belgrade, Serbia; dunjavrsac@gmail.com (D.P.); drmare84@hotmail.com (M.S.); suzyqju@gmail.com (S.D.); dasapreradovic@hotmail.com (D.P.)

**Keywords:** delta, omicron variant COVID-19 infection, maternal, neonatal outcome

## Abstract

*Background and Objectives*: The aim of the present work was to compare the characteristics of delta and omicron variants of COVID-19 infection in pregnant women, the association of infection with comorbidity, clinical manifestation of the disease, type of delivery, and pregnancy outcome. *Material and Methods*: The study was designed as an observational, retrospective study of a single center. The analysis included the cohort of women who had SARS-CoV-2 infection during pregnancy and/or childbirth in the period from 1 March 2020 to 30 June 2023. *Results*: Out of a total of 675 pregnant women with SARS-CoV-2 infection, 130 gave birth with the delta and 253 with the omicron variant. In our retrospective analysis, pregnant women with both SARS-CoV-2 variants had a mild clinical history in most cases. In the omicron period, a significantly lower incidence of pregnancy loss (*p* < 0.01) and premature birth (*p* = 0.62) admission of mothers and newborns to the intensive care unit (*p* < 0.05) was recorded. *Conclusions*: In our retrospective analysis, pregnant women with COVID-19 infection generally exhibited a milder clinical manifestation with both variants (delta and omicron) of the viral infection. During the delta-dominant period, ten percent of affected pregnant women experienced a severe clinical history. However, during the omicron-dominant period infection, a significantly lower incidence of complications, pregnancy loss, preterm delivery, and admission of mothers and neonates to the intensive care unit was recorded. This can be partly explained by the greater presence of pregnant women with natural or induced vaccine immunity.

## 1. Introduction

COVID-19, an infectious disease caused by SARS-CoV-2, was declared a global pandemic by the WHO in March 2020 [1]. COVID-19 has infected more than 500 million people and killed more than 6 million worldwide, causing disruptions to employment, education, food security, and essential health services [2].

Pregnant women in the SARS-CoV-2 pandemic were categorized as high-risk because of additional risks of vertical transmission of infection to the fetus, with unknown consequences for both the fetus and the outcome of the pregnancy. A change in pregnancy protocols was observed, with a significant reduction in the number of invasive prenatal procedures [3]. So far, there is little evidence of vertical transmission from mother to fetus, but there is knowledge that COVID-19 affects the course of pregnancy even in the case of asymptomatic maternal infection [4,5,6].

The COVID-19 infection of the delta variant was registered in the Republic of Serbia on 6 March 2020, and the healthcare system of Serbia was reorganized under pandemic conditions. Many studies have shown that the obstetric population was indeed at increased risk for severe illness associated with COVID-19, with a greater risk of preterm birth and a higher risk of giving birth to infants with a low Apgar score and intrauterine growth restriction as well as infants admitted to neonatal intensive care [7,8,9,10,11]. The vulnerability of pregnant women, fetuses, and newborns to COVID-19 infection was a consequence of physiological, immunological, and cardiovascular changes during pregnancy [7,8,9]. Meta-analyses and observational studies have helped elucidate the problem of COVID-19 infection, emphasizing that the severity of symptoms of SARS-CoV-2 infection (asymptomatic, mild, moderate, and severe infection) is not correlated with adverse pregnancy outcomes [9,10,11,12,13]. The Martinez-Portilla study pointed out that there is no increased susceptibility of pregnant women to SARS-CoV-2 compared to the general population [14]. Despite numerous observational studies, SARS-CoV-2 infection during pregnancy is associated with an increased risk of complications for both the pregnant woman (admission to intensive care and preterm delivery) and the fetus (stillbirth and neonatal mortality) [9,10,11,12]. Since 23 December 2021, the omicron variant of SARS-CoV-2 has also appeared in Serbia. Observational studies indicated that the delta variant (B.1.617.2) was associated with more severe clinical manifestations in pregnancy and worse outcomes; while the omicron variant (B.1.1.529) was found to be milder in terms of clinical manifestations and outcomes in pregnancy, the virulence itself was found to be increased [9,10,11,12,13,14,15,16,17].

The aim of our observational, retrospective, single-center analysis was to compare the characteristics of delta and omicron variants of COVID-19 infection in pregnant women, risk factors, clinical manifestations (symptoms, laboratory), type of delivery, and maternal and perinatal outcomes.

## 2. Material and Methods

The study was designed as an observational, retrospective, single-center study, and the analysis included a cohort of women who had SARS-CoV-2 infection during pregnancy and/or childbirth between 1 March 2020 and 30 June 2023. When the pandemic was lifted in the Republic of Serbia (WHO lifted the SARS-CoV-2 pandemic on 5 May 2023), we defined two exposure periods in the study indicating the times during which the delta or omicron variants were dominant. Based on the status of S gene-positive RT-PCR samples taken from the general population in Serbia (available for 95% of RT-PCR tests during the study period), we defined the delta period from 1 March 2020 to 23 December 2021 (>50% of samples were positive for the S gene) and the omicron period from 23 December 2021 to 30 June 2023 (>50% of samples were negative for the S gene). We chose a threshold of 50% to define dominance so that the results of our study could be compared with the results of other studies. We tested the impact of using this threshold in a pre-defined analysis of infection sensitivity during periods when 90% of SARS-CoV-2 infections were positive for the S gene (delta; 1 March 2020–23 December 2021) or after the point when 90% of SARS-CoV-2 infections became negative for the S gene again (omicron; 23 December 2021–30 June 2023). We were unable to directly use virus sequencing data for our analyses.

COVID-19 infection during pregnancy was generally defined as an infection diagnosed at any time from the conception date to the end of pregnancy. All examined pregnant women who came to the clinic due to accompanying diseases during pregnancy and/or the beginning of childbirth and the infection of COVID-19 were diagnosed during admission to hospital treatment. In proving COVID-19 infection, we used an antigen test—a rapid diagnostic test for detecting antigens (ag-RDT), which is based on a swab from the nasopharyngeal cavity. In cases of a negative rapid antigen test but with symptoms of infection, a molecular genetic PCR test for the SARS-CoV-19 virus was performed. Both variants of COVID-19 infection (delta and omicron variants) were confirmed from nasopharyngeal swabs, and they were sensitive to the same antigen test or molecular genetic PCR test for the SARS-CoV-19 virus.

Pregnant women with infections were divided into groups: mild infection and severe clinical symptoms. Mild infection where pregnant women who had a positive SARS-CoV-2 antigen test with symptoms such as fever, cough, myalgia, anosmia without dyspnea, shortness of breath, and/or abnormal respiratory sounds. Severe clinical symptoms were indicated in pregnant women who had a positive SARS-CoV-2 antigen test and symptoms such as high fever, cough, fatigue, and pneumonia, which can lead to the development of acute respiratory syndrome (SARS) [18,19].

During the pregnancy, we monitored and compared the mother’s age, gestational week at the time of infection, symptoms of the infection, severity of the clinical condition, vaccination status, risk factors, duration of pregnancy, type of delivery, method of pregnancy termination, and perinatal outcome. Vaccination status at the time of infection was categorized as follows: unvaccinated, one dose (one dose of vaccination > 21 days before the date of infection or two doses with the second dose ≤ 14 days before the date of infection), or two doses or more (two doses with the second dose > 14 days before infection). We also analyzed and monitored neonates until the end of the perinatal period (7 postpartum days). Then, we looked at birth weight and score (Apgar score), length of stay in the intensive care unit, and positive nasopharyngeal swabs for SARS-CoV-2. 

Statistical processing and analysis were performed using SPSS version 24 (Statistical Package for the Social Sciences, IBM SPSS Statistics software version 24.0 for Windows, Armonk, NI, USA). Statistical analysis was performed by calculating means and standard deviations, H2 (chi-square test), and ANOVA test for samples.

## 3. Results

From 1 March 2020 to 30 June 2023, a total of 675 pregnant women with SARS-CoV-2 infection were hospitalized at the “Narodni Front” Gynecology and Obstetrics Clinic. There were 322 cases of the delta variant of the SARS-CoV-19 infection and 353 cases of the omicron variant among pregnant women. Of the total number of infected pregnant women, 130 gave birth with the delta variant and 253 with the omicron variant (Figure 1).

A total of 78 patients lost their pregnancies during the delta variant, and 29 pregnant women lost their pregnancies during the omicron variant. A total of 96 women infected during the omicron-dominant period were previously infected during the delta-dominant period. The demographic characteristics of pregnant women infected with SARS-CoV-2 were similar during both periods of COVID-19 infection (Table 1). In both periods of COVID-19, the largest number of pregnant women had the infection in the second trimester of pregnancy. In the delta-dominant period, as many as 52% of all pregnant women were unvaccinated or vaccinated with one dose (*p* < 0.05; Table 1). During the omicron period of infection, 60% of all pregnant women were vaccinated, and 52% received two or more doses (Table 1). The majority of pregnant women in both COVID-19 periods gave birth vaginally (*p* < 0.05; Table 1). The rate of preterm delivery was 13% in the delta and 11% in the omicron period (*p* = 0.62; Table 1).

Data analysis showed that the incidence of mild COVID-19 clinical manifestations was lower in the omicron variant group compared to the delta variant group (177/353 vs. 67/322, *p* < 0.01). The incidence of severe COVID-19 was lower in the omicron variant group compared to the delta variant group (8/353 vs. 33/322, *p* < 0.01; Table 2).

The most common symptoms of COVID-19 infection for both types were headache, followed by loss of smell and taste, asthenia, and myalgia. A significantly higher number of pregnant women in the delta period had severe clinical manifestations and continued medical treatment after delivery in the intensive care unit (Table 2). Severe clinical manifestations (COVID pneumonia with oxygen saturation < 90%; signs of respiratory failure) were observed in 33 deliveries, and further treatments were performed in the intensive care unit and COVID-19 hospitals. COVID-19 hospitals were required for 13 patients with the delta variant and 8 with the omicron variant (*p* < 0.01; Table 2). Only one patient had a fatal outcome within the first 24 h after vaginal delivery, who was infected with the omicron variant of SARS-CoV-19 (an autopsy showed severe bilateral COVID pneumonia). In laboratory analyses of pregnant women with COVID-19 infection, we detected a significant statistical difference between the two subtypes of COVID-19 infection as well as in the length of hospital treatment (*p* < 0.05; Table 2).

During both SARS-CoV-2 infection periods (delta and omicron variants), pregnant women with comorbidities, specifically gestational hypertension or preeclampsia and diabetes, were most affected (Table 3).

Analyzing the characteristics of newborns of patients with COVID-19 infection, we noticed that there was no significant statistical difference in the body weight of newborns at birth and the Apgar score. The only significant statistical difference between delta and omicron variants (*p* < 0.05; Table 4) was in the length of stay in the neonatal intensive care unit.

## 4. Discussion

Methodologically, we divided the entire period of SARS-CoV-2 infection based on the status of S gene-positive RT-PCR samples taken from the general population of Serbia into two periods: the delta-dominance period and the omicron variant-dominant period of SARS-CoV-2. This study is a retrospective analysis of pregnancy outcomes, the impact of comorbidities, type of delivery, and neonatal characteristics concerning two variants of SARS-CoV-2 infection. Based on the analyzed data, we did not find significant differences in demographic characteristics but noticed that around 60% of all pregnant women had an infection in the third trimester with both variants of the SARS-CoV-2 virus. The incidence of infection in the first and second trimesters as well as in the third trimester did not show statistically significant differences between the mentioned two variants of the SARS-CoV-2 virus, which is consistent with a study from Scotland [20]. Observing the vaccination rate, we noticed a significant statistical difference between the two periods of COVID-19 infection. In the delta-dominant period, 52% of pregnant women were not vaccinated, and 38% received only the first dose of the vaccine. In the omicron period, as many as 60% of pregnant women were vaccinated—52% with two or more doses of the vaccine—and 89 pregnant women had acquired immunity (previous delta infection of SARS-CoV-2). Such a high rate of vaccinated pregnant women in the omicron period as well as those with acquired immunity led to a milder clinical manifestation of the disease. The high vaccination rate of pregnant women (over 60%) with milder clinical manifestations of COVID-19 infection was highlighted in the Scottish study [20]. A milder clinical manifestation of COVID-19 infection in pregnant women during the omicron period manifested as loss of smell and taste, headache, asthenia, myalgia, cough, and diarrhea. The omicron COVID-19 infection more frequently had a mild clinical history, which can be partly attributed to previous vaccination of pregnant women (60% vaccinated pregnant women in the omicron period, of which even 52% received two doses of the vaccine). The study by Santos and colleagues focused only on the manifestation of infection and outcome of pregnancies with mild COVID-19 infection in Brazil [21]. In Brazil, the most common infection symptom was headache, followed by loss of taste and smell, and then asthenia [21]. In studies conducted in the USA, the most common symptoms were cough and headache, while loss of smell and taste was recorded in 81% of all respondents [12,22]. In the discussion, the authors agreed that the present difference in dominant symptomatology can be associated with different variants of COVID-19 infection, although in our retrospective analysis, the symptom of loss of smell and taste was dominant in both variants of COVID-19 infection. A meta-analysis published in 2020 mentioned elevated temperature as the dominant symptom, followed by dyspnea, cough, and myalgia [13]. A severe clinical history of COVID-19 infection was observed in 10.2% of all pregnant women with the delta variant compared to eight patients (3.2%) with the omicron variant. The requirement for intensive care unit (ICU) admission was 4% for patients with the delta variant and 2% for those with the omicron variant of COVID-19 infection. Similar results were reported by Stock [20], where severe infection was more frequently associated with the delta variant compared to omicron, which they attributed to the high vaccination rate with at least two doses of vaccine during the omicron period (48% in Scotland and 52% in our study). With prior infection during the delta period and the development of immunity, even 89 pregnant women in our study had both variants: the delta and then omicron variant of COVID-19 infection [20]. Studies by Seasely and Savasi indicated that severe clinical manifestation during the omicron-dominant period of COVID-19 infection was observed only in unvaccinated patients [23,24].

Delta infection of pregnant women was characterized by high C-reactive protein and high D-dimer but significantly lower lymphocytes compared to omicron infection. Platelet and transaminase values were also significantly higher in delta infection compared to omicron, but this can be related to the comorbidity of pregnant women affected in the delta period of infection. Shulman, in his retrospective study, compared the values of D-dimer in delta and omicron variants of the infection, and he pointed out that the significantly higher D-dimer in patients with the delta variant caused devastating thromboembolic effects in infected individuals [25]. Studies by Pereira and Talmac pointed out that high values of D-dimer and C-reactive protein as well as lymphopenia in SARS-CoV-2 infection are significantly associated with a severe form of the disease and poor outcome [26,27].

The severity of the clinical manifestation was influenced by maternal age, risk factors, and obesity, along with pregnancy-related conditions such as preeclampsia and gestational diabetes [13,28].

In our study, the delta variant of infection occurred in 71% of pregnant women with comorbidities, most commonly in women with hypertension/preeclampsia, diabetes, or diagnosed with one of the hereditary thrombophilias. In our population, the incidence of infection with the omicron variant was diagnosed in 24.4% of pregnant women with comorbidities. Brazilian and Scottish studies [20,21,29] have emphasized that SARS-CoV-2 infection is more common in pregnant women with comorbidities, chronic and gestational hypertension, diabetes, thyroid disorders, obese women, and women with HIV. By comparing the characteristics of the clinical manifestations of delta (severe clinical picture 10.2%; ICU 4%) to omicron infection (3.2%; ICU 2%), a significant statistical difference was noted. The omicron variant of infections in our study manifested itself in 96.8% of all cases with an asymptomatic/mild clinical picture, and complications as well as admission to the intensive care unit (ICU) were less frequent. Similar observations were made by Stock, who recorded a severe clinical manifestation in the omicron-dominant period of only 0.3%, and ICU admission was required in only 0.2% of delivery patients [20]. The authors emphasized that no maternal deaths were recorded with the delta or omicron variant of viral infection within 28 days postpartum [20,30,31]. In our study, no deaths were recorded in the delta-dominant period, but in the omicron period, there was a death in the first 24 h after delivery due to fulminant pneumonia in the parturient.

Looking at the rate of preterm deliveries, the average weight of neonates, and Apgar scores at birth between the two variants of SARS-CoV-2, no statistically significant difference was found. A significant statistical difference was observed in the length of neonatal stay in the ICU during the delta-dominant period compared to neonates in the omicron-dominant period (16.1/4.0%). Similar results to ours were reported by the Scottish study as well as the study by Adhikari and colleagues from the USA [16,21]. In our study, in the delta-dominant period, 44% of all births were completed by cesarean section. However, with the duration of the pandemic, the appearance of the omicron variant (a milder clinical history) as well as the improved skills of health personnel in managing the delivery of infected women resulted in a decrease in the incidence of cesarean section. Consequentially, in the omicron-dominant period, 65% of all deliveries were completed vaginally (*p* < 0.05). The study by Norman and colleagues [32] from Sweden and Norway highlighted that over 80% of deliveries in both periods were vaginal, and the percentage of neonates in the ICU was around 10.2% [32]. The study highlighted that there was no increased risk of neonatal adverse outcomes in pregnant women vaccinated during pregnancy [32]. Comparing pre-pandemic neonatal mortality in Sweden (1.3/1000) with pandemic-increased neonatal mortality (1.6–2.0/1000), the authors associated it with the lack of vaccination and poor socioeconomic status in these pregnant women [32]. In our study, there was a significant statistical difference (16/8%) in the delta-dominant period compared to the omicron period in the analysis of lost pregnancies, which can also be linked to the rate of unvaccinated pregnant women in the delta period. An increased rate of lost pregnancies due to COVID-19 infection was registered in the Swedish study by Magnus and his colleagues [33].

The limitations of our study should also be noted. First, we did not have sequencing for all samples, so we used the dominant variant of the wider population at the time of SARS-CoV-2 infection to identify the viral variant. Second, we limited the analysis to the perinatal period, which includes the period from 22 weeks of gestation to 7 days after birth, so we could not monitor neonatal outcomes beyond that period. Third, the sample studied is relatively small, as it represents a sample from one center.

## 5. Conclusions

In our retrospective analysis, pregnant women with COVID-19 infection generally exhibited a milder clinical manifestation with both variants (delta and omicron) of the viral infection. During the delta-dominant period, ten percent of affected pregnant women experienced a severe clinical history. However, during the omicron-dominant period of SARS-CoV-2 infection in pregnancy, a significantly lower incidence of complications, pregnancy loss, preterm delivery, and admission of mothers and neonates to the intensive care unit was recorded compared to the delta-dominant period. This can be partly explained by the greater presence of pregnant women with natural or induced vaccine immunity.

## Figures and Tables

**Figure 1 medicina-60-00935-f001:**
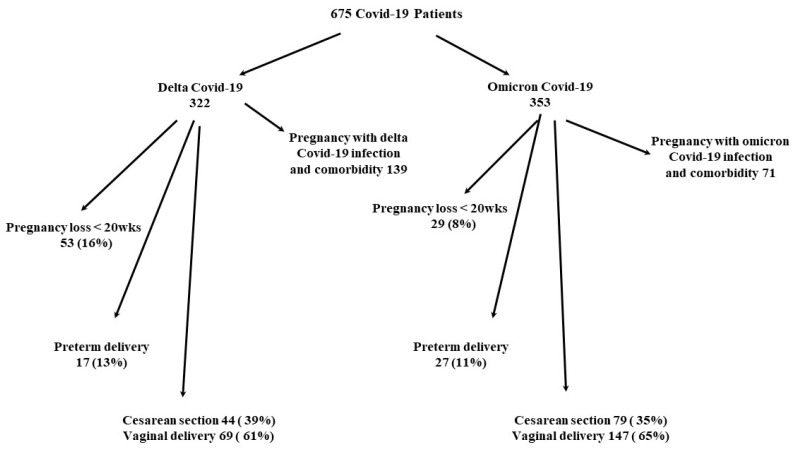
Flow chart of the study population.

**Table 1 medicina-60-00935-t001:** Clinical characteristics of a pregnant woman with delta and omicron COVID-19 infection.

Characteristics of the Pregnant Woman	Delta COVID-19	Omicron COVID-19	*p*-Value
Mean maternal age (X ± SD) (min/max)	30.66 ± 2.02 (17/42)	32.30 ± 1.78 (18/53)	^b^ 0.621
Primigravida (N/%)	151/46.9	170/48.2	^a^ 0.67
Infection to first trimester (<13 + 6 weeks) (N/%)	32/10	56 /16	^a^ 0.70
Infection to the second trimester (14–27 + 6 weeks) (N/%)	111/34.6	77/22	^a^ 0.67
Infection to third trimester (>28 weeks) (N/%)	178/55.4	218/62	^a^ 0.67
Unvaccinated (N/%)	167/52	141/40	^a^ <0.05
One dose	122/38	28 /8	^a^ <0.05
Two or more doses	31/10	184/52	^a^ <0.01
Mean weeks of delivery (X ± SD) (min/max)	38.46 ± 1.02 (28/41)	37.68 ± 2.22 (25/42)	^b^ 0.88
Spontaneous preterm birth (<37 weeks)	17/13	27/11	^a^ 0.62
Pregnancy loss (<20 weeks)	53/16	29/8	^a^ <0.01
Cesarean section (N/%)	44/39	79/35	^a^ <0.05
Vaginal delivery (N/%)	69/61	147/65	^a^ <0.05

Abbreviations: X ± SD, mean value standard deviation; Test: ^a^ χ^2^, chi-square test; ^b^ ANOVA test; *p* = statistical significance.

**Table 2 medicina-60-00935-t002:** Rates of clinical manifestation of delta and omicron COVID-19 infection.

Clinical Manifestation of COVID-19 Infection		Delta COVID-19 (N/%)	Omicron COVID-19 (N/%)	*p*-Value
Without symptoms COVID-19		30/23	68/26.8	^a^ <0.05
Mild COVID-19 Clinical manifestation		67/51.5	177/70.0	^a^ <0.01
Mild COVID-19	Loss of smell/taste	88.7	90	^a^ 0.76
Mild COVID-19	Headache	85.4	78.7	^a^ 0.78
Mild COVID-19	Asthenia	76	74	^b^ 0.81
Mild COVID-19	Myalgia	69	64	^b^ 0.79
Mild COVID-19	Cough	66	59	^a^ 0.9
Mild COVID-19	Diarrhea	34	56	^a^ 0.72
Severe COVID-19		33/10.2	8/3.2	^a^ <0.01
Admission to the intensive care unit (ICU admission)		13/4	5/2	^a^ <0.05
Exitus details (maternal death)		0	1/0.4	/
Average number of days of hospitalization		7.217 ± 0.82	4.711 ± 0.14	^b^ <0.05
Raised white cell count		296/92	240/68	^a^ <0.01
Lymphopenia		309/96	257/73	^a^ <0.01
Thrombocytopenia		116/36	56/16	^a^ <0.05
Abnormal liver function test results		93/29	40/11	^a^ 0.038
Raised C-reactive protein level		87/27	33/9	^a^ <0.01
Raised D-dimer		180/56	144/31	^a^ <0.01

Abbreviations: ICU, intensive care unit; Test: ^a^ χ^2^, chi-square test; ^b^ ANOVA test; *p* = statistical significance.

**Table 3 medicina-60-00935-t003:** Comorbidity in pregnant women with delta and omicron COVID-19 infection.

Comorbidity in Pregnancy	Delta COVID-19 (N/%)	Omicron COVID-19 (N/%)	*p*-Value
HTA gest/Preeclampsia	83/26.1	35/10	^a^ <0.05
HTA chronic	29/9.2	18/5.1	^a^ <0.05
Asthma bronchiale	17/5.4	1/0.4	/
DM gestational/type I	52/16.1	18/5.1	^a^ <0.05
St post CVI	2/0.76	/	/
Hypothyreosis	5/1.5	7/2	^a^ 0.78
Thrombophilia	42/13	7/2	^a^ <0.05
BMI (kg/m^2^) > 30	34	27	^a^ <0.05

Abbreviations: HTA gest, gestational hypertension; HTA chronic, chronic hypertension; DM gestational/type I, diabetes mellitus gestational/type I; St post CVI, St post cardiovascular insults; BMI, body mass index. Test: ^a^ χ^2^, chi-square test.

**Table 4 medicina-60-00935-t004:** Outcomes in pregnant women with delta and omicron COVID-19 infection.

Neonatal Characteristics	Delta	Omicron	*p*-Value
Birth weight (gram)	3305.481 ± 538.642	3185.696 ± 711.321	^b^ 0.72
Mean Apgar score in 5′ (X ± SD)	9.7 ± 0.34	9.5 ± 0.45	^b^ 0.8
Abnormal Apgar score < 7 (N/%)	6/4.6	5/2.0	^a^ 0.67
ICU > 24 h (N/%)	21/16.1	10/4.0	^a^ <0.05
SARS-CoV-19 neonates (N/%)	0	0	/

Abbreviations: ICU, intensive care unit; X ± SD, mean Apgar score ± standard deviation; significant significance. Test: ^a^ χ^2^, chi-square test; ^b^ ANOVA test.

## Data Availability

Original data are available on request.

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
