# Peer review of "Comparison of Perinatal Outcome of Delta and Omicron Variant of COVID-19 Infection—A Retrospective Observational Study"

_medicina, 2024, doi:10.3390/medicina60060935_

Round 1

Reviewer 1 Report

Comments and Suggestions for Authors

Dear authors, thanks for your manuscript. have you also histologic information of the placentas in infected mothers Vs non infected mothers? Have you analyzed the placentas of mothers with loss of pregnancy/still birth/premature delivery? Are your data congruent with the literature in terms of difference between delta and Omicron? Have you also a confirmation test that attested which variant was for every patient?

Thanks

Comments on the Quality of English Language

Dear authors, please revise your manuscript also for the quality of English language. A minor revision is needed.

Author Response

Response to Reviewers for the manuscript ID: medicina-2973837
Response to Reviewer 1:

  • Dear authors, thanks for your manuscript. have you also histologic information of the placentas in infected mothers Vs non infected mothers? Have you analyzed the placentas of mothers with loss of pregnancy/still birth/premature delivery?

Dear reviewer, thank you for your question. Recommendations in the period of the COVID-19pandemic were that infectious material should not be analysed histologically, but with special precautions it should be disposed, so unfortunately we do not have a histological analysis of the placenta of mothers with pregnancy loss/stillbirth/premature birth delivery compared to the non-infected population of pregnant women(reference attached).

  • Pelemiš M, Stevanovic G, Turkulov V, Vucinic V, Matijasevic J, Milosevic B, Milosevic I, Gajovic O, Ladjevic N, Vrbic M, Barac B, Udovicic I, Jankovic R, Mikic D, Antic D, Mujovic N, Djordjevic M, Korac M, Ranin J, Bukarica Lj, Dragovic G. Protokol lecenja pacijenata sa Covid 19. 2022; verzija 13:1-17.

  • Are your data congruent with the literature in terms of difference between delta and Omicron?

Dear reviewer, thank you for your question.Yes, our results are in agreement with studies from Australia, Scotland, Brazil, USA, North Africa, Sweden and Norway, which we discussed in the discussion section. (references attached)

  • Pettirosso E, Giles M, Cole S, Rees M, COVID-19 and pregnancy: a review of clinical characteristics, obstetric outcomes, and vertical transmission. Austral N Z J Obstet Gynaecol 2020; 60: 640–659.
  • Stock* SS, Moore* E, Calvert C, Carruthers J, Denny C, Donaghy J, Hillman S, Hopcroft, Leanne Hopkins L SM, Goulding L, Lindsay L, McLaughlin T, Taylor B, Auyeung B, Vittal Katikireddi S, McCowan C, Sir Ritchie LD, Rudan I, Simpson CR, Robertson C, Sir Sheikh A, Wood R.Pregnancy outcomes after SARS-CoV-2 infection in periods dominated by delta and omicron variants in Scotland: a population-based cohort study. Lancet Respir Med 2022;10:1129–36.
  • Santos CAD, Fonseca Filho GG, Alves MM, Macedo EYR,. de A. Pontes AA,, Paula AP, Barreto CTR, Zeneide FN. Maternal and Neonatal Outcomes Associated with Mild COVID-19 Infection in an Obstetric Cohort in Brazil. Am. J. Trop. Med. Hyg., 107(5), 2022, pp. 1060–1065.
  • Breslin N, Baptiste C, Gyamfi-Bannerman C, et al. COVID-19 infection among asymptomatic and symptomatic pregnant women: two weeks of confirmed presentations to an affiliated pair of New York City hospitals. Am J Obstet Gynecol MFM 2020;2:100118.
  • Seasely AR, Blanchard CT, Arora N, et al. Maternal and perinatal outcomes associated with the omicron variant of severe acute respiratory syndrome coronavirus 2 (SARS-CoV-2) infection.Obstet Gynecol 2022; 140: 262–65.
  • Savasi VM, Parisi F, Patanè L, et al. Clinical findings and disease severity in hospitalized

   pregnant women with coronavirus disease 2019 (COVID-19). Obstet Gynecol 2020;136:252-8.

  • Schulman AH, Jacobson B, Segal BM, Khan A, Trusler J, Earlam L, Shemesh G. D-dimers in omicron versus delta: A retrospective analysis. Southern African J of Infectious Disease. 2022;37:1-6.
  • Pereira A, Melguizo SC, Adrien M, Fuentes L, Marin E, Perez-Medina T. Clinical course of coronavirus disease-2019 in pregnancy. Acta Obstet Gynecol Scand. 2020;99:839–847.
  • Norman M, Magnus MC, Soderling J. Neonatal outcomes after Covid-19 vaccination in

   pregnancy. JAMA 2024; 331(5):396-407.

  • Magnus MC, Ortqvist AK, Kjaer US. Infection with Sars-CoV-2 during pregnancy and risk of stillbirth: a Scandinavian registry study. BMJ 2022;378:e071416.

  • Have you also a confirmation test that attested which variant was for every patient?

Dear reviewer, thank you for your question, No, we were not able to check the COVID-19variant of the virus in every patient, which we stated in the methodology (attachment).

“We defined two exposure periods in the study indicating the times during which the delta or omicron variants were dominant. Based on the status of S gene-positive RT-PCR samples taken from the general population in Serbia (available for 95% of RT-PCR tests during the study period), we defined the delta period from March 1, 2020, to December 23, 2021 (>50% of samples were positive for the S gene) and the omicron period from December 23,2021, to June 30, 2023 ( >50% of samples were negative for the S gene). We chose a threshold of 50% to define dominance so that the results of our study could be compared with the results of other studies. We tested the impact of using this threshold in a predefined analysis of infection sensitivity during periods when 90% of SARS-CoV-2 infections were positive for the S gene (delta; March 1, 2020 - December 23, 2021) or after the point when 90% of SARS-CoV-2 infections became negative for the S gene again (omicron; December 23, 2021 - June 30, 2023). We were unable to directly use virus sequencing data for our analyses.”

Reviewer 2 Report

Comments and Suggestions for Authors

Abstract:

Results section should be more focused on the effective results of the analysis. Please delete “Demographic characteristics, vaccination rate, type of delivery, rate of premature births as well as the most common clinical history of pregnant women infected with SARS-CoV-2 were similar in both periods”, include the subsequent corrected paragraph “In our retrospective analysis, pregnant women with both SARS-CoV-2 variants had a mild clinical history in most cases. In the omicron period, a significantly lower incidence of pregnancy loss, premature birth, admission of mothers and newborns to the intensive care unit was recorded (INCLUDE STATISTICS WITH χ2 VALUE AND p FOR ALL THESE PARAMETERS).”

Then include a conclusion sentence for example “In our retrospective analysis, pregnant women with Covid-19 infection generally exhibited a milder clinical manifestation with both variants (delta and omicron) of the viral infection. This can be partially ex-plained by the higher presence of women with natural or induced immunity.”

Introduction Page 2 first paragraph: please include also the impact of COVID on the prenatal services (PMID: 34618213) and the impact of COVID on the fetus, even in cases without particular symptoms (PMID: 36677515).

Table 1: Please include the χ2 value in the Table for the Chi-square test analyses. There are statistically significant results that should be included also in the manuscript with the relative analyses (decreased incidence of pregnancy loss and increased incidence of CS).

Correct “Tbl2” and “Tbl3” at pages 4-5-6.

Table 2 and 3. Please include the χ2 value in the Table for the Chi-square test analyses. Again, the significant results are only mentioned in the main text: Authors should explain to the readers the results of the statistical analysis in a very extensive way. For example: “Data analysis showed that the incidence of mild COVID-19 clinical manifestations was decreased in the omicron variant group compared to the delta variant group (177/353 vs 67/322, χ2 …., p <0.01; the incidence of severe COVID-19 was decreased in the omicron variant group compared to the delta variant group (8/353 vs 33/322, χ2 …., p <0.01)” etc….

Correct Tbl4 at page 7

Discussion: a recent study on a large cohort of COVID-19 infected pregnant women showed also a decrease in the temporal trends of CS from the beginning of the pandemic through the subsequent years. This could be due to the different severity of COVID-19 variants but also the the improved skills of healthcare personnel in the management of delivery of infected women . Please include and discuss these data in  the light of your results.

Author Response

Response to Reviewers for the manuscript ID: medicina-2973837
Response to Reviewer 2:

  • Abstract:

Results section should be more focused on the effective results of the analysis. Please delete “Demographic characteristics, vaccination rate, type of delivery, rate of premature births as well as the most common clinical history of pregnant women infected with SARS-CoV-2 were similar in both periods”, include the subsequent corrected paragraph “In our retrospective analysis, pregnant women with both SARS-CoV-2 variants had a  mild clinical history in most cases. In the omicron period, a significantly lower incidence of pregnancy loss, premature birth, admission of mothers and newborns to the intensive care unit was recorded (INCLUDE STATISTICS WITH X2 VALUE AND p FOR ALL THESE PARAMETARS )

Dear reviewer, thank  you for your suggestion.

We agreed with this comment and modified the Resuls section accordingly:

In our retrospective analysis, pregnant women with both SARS-CoV-2 variants had a mild clinical history in most cases. In the omicron period, a significantly lower incidence of pregnancy loss (p <0.01) premature birth (p=62)admission of mothers and newborns to the intensive care unit (p<0.05) was recorded.

Then include a conclusion sentence for example “In our retrospective analysis, pregnant women with Covid-19 infection generally exhibited a milder clinical manifestation with both variants (delta and omicron) of the viral infection. This can be partially ex-plained by the higher presence of women with natural or induced immunity.”

Dear reviewer, thank you for your suggestion. We agreed with this comment and modified the Conclusion section accordingly:

In our retrospective analysis, pregnant women with Covid-19 infection generally exhibited a milder clinical manifestation with both variants (delta and omicron) of the viral infection. This can be partially ex-plained by the higher presence of women with natural or induced immunity. ”

  • Introduction Page 2 first paragraph: please include also the impact of COVID on the prenatal services (PMID: 34618213) and the impact of COVID on the fetus, even in cases without particular symptoms (PMID: 36677515).

Dear reviewer, thank you for your suggestion. We agreed with this comment and modified the Introduction section accordingly:

Pregnant women in a pandemic become a risk category for additional risks of vertical transmission of infection to the fetus, with unknown consequences for both the fetus and the outcome of the pregnancy.

A change in pregnancy protocols has been observed, with a significant reduction in the number of invasive prenatal procedures (3). So far, there is little evidence of vertical transmission from mother to fetus, but there is knowledge that COVID-19 affects the course of pregnancy even in case of asymptomatic maternal infection. (4, 5, 6).

  1. Carbone L, Raffone A, Sarno L, Travaglino A, Saccone G, Gabrielli O, Migliorini S, Sirico A, Genesio R, Castaldo G, Capponi A, Zullo F, Rizzo G, Mariotti GM. Invasive prenatal diagnosis during COVID‐19 pandemic.Archives of Gynecology and Obstetrics 2022; 305:797–801.

  1. Ferrazzi E, Frigerio L, Savasi V et al Vaginal delivery in SARS-CoV-2 infected pregnant women in Northern Italy: a retrospective analysis. BJOG. 2020; 9:1116-21.

  1. Cinnamon M, LaCivita E, Sarno L, Carbone G, DiSomma S, Camaro S, Troisi J, Sirico A, Improda FP, Guida M, Terracciano D, Portella G. Low Interferon-γ Levels in Cord and Peripheral Blood of Pregnant Women Infected with SARS-CoV-2. Microorganisms 2023;11: 223-234.

  1. Vasilescu DI, Rosoga AM, Vasilescu S, Dragomir I, Dima V, Dan AM, Cirstoiu MM. SARS-CoV-2 Infection during Pregnancy Followed by Thalamic Neonatal Stroke-Case Report. Children (Basel). 2023 May 27;10(6):958-968.

  • Table 1: Please include the χ2 value in the Table for the Chi-square test analyses. There are statistically significant results that should be included also in the manuscript with the relative analyses (decreased incidence of pregnancy loss and increased incidence of CS).

Dear reviewer, thank you for your suggestion. We agreed with this comment and removed and put p values of statistical significance in the tables.

  • Correct “Tbl2” and “Tbl3” at pages 4-5-6.

Dear reviewer, thank you for your suggestion. We agreed with this comment and removed the abbreviations accordingly:Table 2. Table 3. Table 4.

  • Table 2 and 3. Please include the χ2 value in the Table for the Chi-square test analyses. Again, the significant results are only mentioned in the main text: Authors should explain to the readers the results of the statistical analysis in a very extensive way. For example: “Data analysis showed that the incidence of mild COVID-19 clinical manifestations was decreased in the omicron variant group compared to the delta variant group (177/353 vs 67/322, χ2 …., p <0.01; the incidence of severe COVID-19 was decreased in the omicron variant group compared to the delta variant group (8/353 vs 33/322, χ2 …., p <0.01)” etc….

Dear reviewer, thank you for your suggestion. We agreed with this comment and removed and put  p values of statistical significance in the table and in the Results section:

Data analysis showed that the incidence of mild COVID-19 clinical manifestations was decreased in the omicron variant group compared to the delta variant group (177/353 vs 67/322,  p <0.01); the incidence of severe COVID-19 was decreased in the omicron variant group compared to the delta variant group (8/353 vs 33/322, p <0.01; Table  2).

  • Correct Tbl4 at page 7

Dear reviewer, thank you for your suggestion. We agreed with this comment and removed the abbreviations accordingly: Table 4. at the page 7.

  • Discussion: a recent study on a large cohort of COVID-19 infected pregnant women showed also a decrease in the temporal trends of CS from the beginning of the pandemic through the subsequent years. This could be due to the different severity of COVID-19 variants but also the the improved skills of healthcare personnel in the management of delivery of infected women . Please include and discuss these data in  the light of your results.

Dear reviewer, thank you for your suggestion. We agreed with this comment and and modified the Discussion section accordingly:

In our study, in the delta, dominant period, 44% of all births were completed by caesarean section, but with the duration of the pandemic, the appearance of the omicron variant, a milder clinical history, as well as the  improved skills of health personnel in managing the delivery of infected women, the incidence of caesarean section falls,  so that in the omicron dominant period, 65% of all deliveries were completed vaginally (p<0.039).

Reviewer 3 Report

Comments and Suggestions for Authors

This is a very up-to-date paper about SARS-COV-2 infection in pregnancy.

I think it would be best for you to show the meaning of the abbreviations you used under each table. For example, NS means non-specified. Or something else?

Also, explain why the p-value was not calculated for each correlation with different factors.

I published an article about SARS-COV-2 infection in pregnancy and it may help you

Vasilescu DI, Rosoga AM, Vasilescu S, Dragomir I, Dima V, Dan AM, Cirstoiu MM. SARS-CoV-2 Infection during Pregnancy Followed by Thalamic Neonatal Stroke-Case Report. Children (Basel). 2023 May 27;10(6):958. doi: 10.3390/children10060958. PMID: 37371190; PMCID: PMC10297221.

Please verify the spelling in your manuscript.

Comments on the Quality of English Language

Minor editing of English language required

Author Response

Response to Reviewers for the manuscript ID: medicina-2973837
Response to Reviewer 3:

  • This is a very up-to-date paper about SARS-COV-2 infection in pregnancy.

I think it would be best for you to show the meaning of the abbreviations you used under each table. For example, NS means non-specified. Or something else?

Dear reviewer, thank you for your suggestion. We have put an explanation of the abbreviations below each table. For example a Table 1.

Table 1. Clinical characteristics of a pregnant woman with delta and omicron Covid-19 infection

Abbreviations: X±SD mean value standard deviation; NS no statistically significant difference;  a χ2—chi-square test; b ANOVA test Test; p = statistical significance.

  • Also, explain why the p-value was not calculated for each correlation with different factors.

Dear reviewer, thank you for your question. We performed a statistical significance test for each variable.Statistical analysis was performed by calculating means and standard deviations, H2 (chi-square test) and ANOVA test Test for samples. We have entered the p value of statistical significance into the tables and besides put an explanation of the abbreviations below each table.

  • I published an article about SARS-COV-2 infection in pregnancy and it may help you

Vasilescu DI, Rosoga AM, Vasilescu S, Dragomir I, Dima V, Dan AM, Cirstoiu MM. SARS-CoV-2 Infection during Pregnancy Followed by Thalamic Neonatal Stroke-Case Report. Children (Basel). 2023 May 27;10(6):958. doi: 10.3390/children10060958. PMID: 37371190; PMCID: PMC10297221.

Dear reviewer, thank you for your suggestion. I quoted your study in the Introduction section.

  • Please verify the spelling in your manuscript.

Dear reviewer, thank you for your suggestion.I'll check the spelling in my paper.

Round 2

Reviewer 1 Report

Comments and Suggestions for Authors

Dear Authors, I appreciate your efforts but without histological exam your conclusion lacks of novelty. Thanks

Author Response

Dear editor, thank you for your suggestion.

  1. Title can be condensed and improved.

Dear editor, thank you for your suggestion. We agreed with this comment and modified the Title : Comparison of  perinatal outcome of Delta and Omicron variant of Covid-19 infection - A retrospective observational study

Conclusion should be dran from the study findings. We agreed with this comment and modified the Conclusion section:

Conclusion: In our retrospective analysis, pregnant women with Covid-19 infection generally exhibited a milder clinical manifestation with both variants (delta and omicron) of the viral infection. During the delta-dominant period, ten percent of affected pregnant women experienced a severe clinical history. However, during the omicron-dominant period infection, a significantly lower incidence of complications, pregnancy loss, preterm delivery, and admission of mothers and neonates to the intensive care unit was recorded. This can be partially explained by the higher presence of women with natural or induced immunity.

  1. Tables can be condensed. Title of Tables to be on the top and expansion for

abbreviations and other details below the Tables

Dear editor, thank you for your suggestion. We agreed with this comment and modified the Title of Tables and abbreviations :

Table 3. Comorbidity in pregnant women with delta and omicron  Covid-19 infection. 

Abbreviations:HTA gest Gestational Hypertension; HTA chronic Chronic hypertension; DM gestatione/ tip I Diabetes mellitus gestations /type  I; St post CVI.  St post cardio-vascular insults; BMI. Body Mass Index.a χ2—chi-square test; b ANOVA test Test; p = statistical significance.

Dear editor, thank you for your suggestion. We modified (summarized) Table 1. I removed the rows: childbirth with the maximum and minimum week of pregnancy; and Vaccinated with three or more doses, and now it is with two or more doses

  1. It should be 'Material and methods' and not 'Materials'

 Dear editor, thank you for your suggestion. We agreed with this comment and modified the Materials and Methods.

  1. Please check the statistical significance once again since some numbers

appear to be very small.

Dear editor, thank you for your suggestion, I checked the statistical values in the tables.

5.References to be listed uniformly as per journal style.

Dear editor, thank you for the suggestion, the references have been corrected according to the Medicina Journal propositions.

Kind regards,

Natasa Karadzov Orlic

Cover Letter

Medicina

            Dear Sir,

            We submit the original regular article « Comparison of  perinatal outcome of Delta and Omicron variant of Covid-19 infection - A retrospective observational study» to the Medicina, for potential publication.

We state that this article has not been published before and it is not under consideration for potential publication enywhere else.

All authors contributed to the preparation of this article and read it and approved it for the submission to the Journal.

All authors have nothing to disclose regarding this article.

Sincerely,

Natasa Karadzov Orlic MD, PhD

High Risk Pregnancy Department

Ob/Gyn Clinic «Narodni front»

School of Medicine, University of Belgrade,

Kraljice Natalije Street 62.
